# FAWPA: A FAW Attack Protection Algorithm Based on the Behavior of Blockchain Miners

**DOI:** 10.3390/s22135032

**Published:** 2022-07-04

**Authors:** Yang Zhang, Xiaowen Lv, Yourong Chen, Tiaojuan Ren, Changchun Yang, Meng Han

**Affiliations:** 1College of Information Science and Technology, Zhejiang Shuren University, Hangzhou 310015, China; 20081203049@smail.cczu.edu.cn (Y.Z.); xwl1129@zjsru.edu.cn (X.L.); rentj@zjsru.edu.cn (T.R.); 2School of Computer and Artificial Intelligence, Changzhou University, Changzhou 213164, China; ycc@cczu.edu.cn; 3Binjiang Institute, Zhejiang University, Hangzhou 310053, China; mhan@zju.edu.cn

**Keywords:** blockchain, FAW attack, proof of work, malicious miner detection, industrial internet

## Abstract

Blockchain has become one of the key techniques for the security of the industrial internet. However, the blockchain is vulnerable to FAW (Fork after Withholding) attacks. To protect the industrial internet from FAW attacks, this paper proposes a novel FAW attack protection algorithm (FAWPA) based on the behavior of blockchain miners. Firstly, FAWPA performs miner data preprocessing based on the behavior of the miners. Then, FAWPA proposes a behavioral reward and punishment mechanism and a credit scoring model to obtain cumulative credit value with the processed data. Moreover, we propose a miner’s credit classification mechanism based on fuzzy C-means (FCM), which combines the improved Aquila optimizer (AO) with strong solving ability. That is, FAWPA combines the miner’s accumulated credit value and multiple attack features as the basis for classification, and optimizes cluster center selection by simulating Aquila’s predation behavior. It can improve the solution update mechanism in different optimization stages. FAWPA can realize the rapid classification of miners’ credit levels by improving the speed of identifying malicious miners. To evaluate the protective effect of the target mining pool, FAWPA finally establishes a mining pool and miner revenue model under FAW attack. The simulation results show that FAWPA can thoroughly and efficiently detect malicious miners in the target mining pool. FAWPA also improves the recall rate and precision rate of malicious miner detection, and it improves the cumulative revenue of the target mining pool. The proposed algorithm performs better than ND, RSCM, AWRS, and ICRDS.

## 1. Introduction

The industrial internet applies intelligent terminals with sensing capabilities, ubiquitous mobile computing models and real-time information communication to all aspects of industrial production, and realizes the digital, networked, and intelligent transformation of intelligent industry [1]. In recent years, the industrial internet has grown exponentially. According to data from the Ministry of Industry and Information Technology Research Institute, the scale of China’s industrial internet industry has exceeded the trillion yuan mark. The industrial internet is still in a critical period of development. The industrial internet systems should have the characteristics of solid real-time performance, a high degree of automation, high security, and information interconnection. However, the current systems are often traditional centralized systems that face challenges such as the poor computing power of conventional centralized architecture and weak security protection capabilities. As an emerging internet technology, blockchain is showing an explosive growth trend. Due to the decentralized, collective maintenance and tamper resistance of the blockchain, it can meet the characteristics of security storage, privacy protection, and efficient information updating for the industrial internet. Blockchain is also considered the fifth disruptive innovation in the computing paradigm after mainframes [2], personal computers, the internet, mobile networks, and social networks. The blockchain is widely used in finance, the internet of things, data services and other fields and has broad value prospects [3]. Therefore, it has become a trend of applying blockchain technology to the industrial internet. However, in building the industrial internet blockchain system, malicious miners will attack the blockchain system through illegal behaviors, posing a challenge to the security of the industrial internet.

Most consensus algorithms based on Proof of Work (PoW) are used in blockchain applications to achieve block consensus among different miners [4,5]. The PoW consensus algorithm determines the block by calculating the difficulty value and obtaining the revenues. The greater the computing power of miners, the more likely they are to obtain coin rewards and bookkeeping rights. Considering the limitation of computing power and other factors, most of the miners in the network cannot obtain a stable revenue. Then, miners will choose to join the mining pool for cooperative mining [6]. However, in the actual process, there have been many attack methods against mining pools, such as the block withholding attack (BWH), the selfish mining attack, and the Fork after Withholding (FAW) attack [7]. A block withholding attack means that the attacker only sends partial proof of work (PPoW) to the mining pool manager and discards it when the full proof of work (FpoW) is generated [8,9]. A selfish mining attack means that the attacker causes network forks by continuously publishing multiple FPoWs [10]. A FAW attack means that the attacker “throws away” the previously discovered FPoW like BWH. When other miners outside the pool discover new blocks, the attacker will use selfish mining to submit the previously discovered FPoW to cause a fork [11]. Therefore, the FAW attack is a new attack method that combines the block withholding attack and the selfish mining attack. Under a FAW attack, there are malicious mining pools, target mining pools, and other mining pools in the blockchain network. Among them, the malicious mining pool assigns not only malicious miners to attack the target mining pool, but also retains honest miners for honest mining. If a malicious mining pool successfully mines an FPoW through honest miners, it will broadcast the PoW immediately to obtain block revenue. If the malicious miners assigned by malicious mining pool successfully mine the FPoW in the target mining pool, it is necessary to further judge whether other mining pools have found the FPoW. If other mining pools have not mined the FPoW, the malicious miner will continue to retain the FPoW and not submit it. Otherwise, the malicious miner will immediately submit the FPoW to the target mining pool manager. Then, it competes for the current block revenue, resulting in a network fork. In a word, FAW attack not only help attackers to obtain the revenue of block withholding attacks, but also to obtain the revenue from network forks. Thus, FAW attack is a more threatening attack method for mining pools compared with BWH and selfish mining attacks.

In response to FAW attacks by malicious miners in the blockchain network, a FAW attack protection algorithm is needed to ensure the revenue of honest mining. Through the analysis of related work, some scholars currently propose protection algorithms based on the FAW attack’s block withholding feature or selfish mining feature [12,13,14,15,16], but they still have the following three problems:The current protection algorithms only set a simple protection strategy based on a single attack feature. They do not consider the multi-dimensional behavior characteristics of FAW attacks such as block withholding attacks and selfish mining attacks, and do not conduct reasonable miner credit evaluation.The current protection algorithms difficultly detect malicious miners who carry out FAW attacks effectively, and the detection precision rate is low.The current protection algorithms lack a mining revenue model and difficultly evaluate the effect of protection revenues.

In response to the above problems, we propose a FAW attack protection algorithm (FAWPA) based on the behavior of blockchain miners. The contributions of the paper are as follows:We propose a behavioral reward and punishment mechanism and credit value scoring model. Namely, according to honest mining or FAW attacks on target mining pools by the tasks of malicious mining pools, we extract mining behavior characteristics such as offline times, delay time, and current network forks, and propose reward and punishment mechanisms, including PoW reward and block mining failure punishment. Then, by calculating the cumulative performance value of a miner in block mining, we propose the miner’s credit value scoring model to calculate the current credit value. It can mobilize the enthusiasm of miners to participate in mining as the basis for credit ratings.We propose a miner’s credit classification mechanism based on fuzzy C-means (FCM), which combines the improved Aquila optimizer (AO). Namely, according to the evaluation elements of credit rating classification in miner behavior data, the mechanism considers the distance inside and outside of the cluster, and improves the fitness calculation method. It also introduces multiple weight parameters to improve the distance calculation method. It optimizes cluster center selection by simulating Aquila predation behavior, and improves the solution update mechanism in different optimization stages. Then, mining pool administrators obtain the distribution of malicious miners in the mining pool and give the corresponding revenues distribution weights to the detected malicious miners to improve detection speed.We propose the revenue model of the mining pool under FAW attack and the revenue model of each miner. Namely, we extract mathematical formulas such as the effective computing power, mining cost, and revenue of malicious mining pools. Then. the model calculates the revenue of the target mining pool, and quickly evaluates the malicious miner protection effect of the target mining pool. It is conducive to the rapid simulation verification of the algorithm.

The remainder of the paper is organized as follows. Section 2 details the current attack strategies and corresponding protection methods against FAW attacks. Section 3 details the scene description, principles, and processes of the algorithm. Section 4 describes the implementation process and logic code of the algorithm. Section 5 conducts experimental simulations to compare and analyze algorithms, and Section 6 presents the conclusions.

## 2. Related Work

At present, related references mainly focus on improving the revenue of FAW attacks [17,18] and cannot provide corresponding protection algorithms for the target mining pool under FAW attacks. However, there are a few study results on the protection algorithm for FAW attacks. Some scholars proposed a protection algorithm based on the FAW attack’s block withholding feature or selfish mining feature. For example, Chang et al. [12] proposed a protection algorithm based on silent timestamps. Miners send the randomly generated silent timestamp and PoW to the mining pool manager. The mining pool manager sorts the PoWs of miners according to the received silent timestamps, so that the PoWs submitted are time-sensitive. Therefore, it is difficult for a FAW attacker to cause a network fork, and only the block withholding attack is carried out. Ke et al. [13] proposed a reputation system based on the miner credit mechanism (RSCM). RSCM evaluates the miner’s credit value according to the network fork number caused by the miner. Then, RSCM assists the mining pool manager to eliminate malicious miners who initiate FAW attacks, and ensure the revenue of honest mining in the mining pool. Sarker et al. [14] proposed an anti-withholding reward system (AWRS). AWRS removes the incentive for attackers to carry out FAW attacks by giving more enormous rewards to miners with FPoW. Bag et al. [15] deters attackers by providing additional incentives to miners who actually find blocks, and reduces the reward of BWH attackers who never submit valid blocks to the mining pool. BWH attackers’ revenue is lower than expected, which can achieve a defensive effect. Schrijvers et al. [16] proposed an incentive-compatible revenue distribution on scheme (ICRDS). ICRDS reduces the submitted PPoW by setting a threshold for the number of miners’ PPoW submissions. The deducted reward is given as a special reward to miners who complete the FPOW. However, the above scholars [13,14,15,16] only detect malicious miners based on a single attack feature, but do not consider both the features of block withholding and selfish mining. Therefore, it is impossible to comprehensively and efficiently detect malicious miners who carry out FAW attacks and reduce the revenue of malicious miners.

In conclusion, for FAW attacks on other mining pools or miners, the current algorithms focus on maximizing the attacker’s revenue by optimizing the existing attack strategies. There are a few studies on protection algorithms. Moreover, the current algorithms only set a simple protection algorithm based on a single attack feature, and they do not comprehensively and efficiently detect malicious miners and effectively reduce the revenue of malicious miners from FAW attacks.

## 3. Algorithm Principle

### 3.1. Algorithmic Assumptions and Problem Solving

To address the above problem, we analyze the blockchain network scenario and give the assumptions as follows.

**Assumption** **1.***There are three types of mining pools in the network. Malicious mining pools send not only malicious miners to attack target mining pools, but also retain honest miners for honest mining. The target mining pool and other mining pools only assign honest miners for honest mining*.

**Assumption** **2.***Honest miners can mine by themselves and obtain the mining revenue distributed by their mining pools. Malicious miners carry out FAW attacks on the target mining pool according to the tasks of the malicious mining pool. At the same time, the malicious miners obtain the attack reward distributed by the malicious mining pool and the mining revenue distributed by the target mining pool*.

**Assumption** **3.***Miners in the mining pool use the PoW consensus algorithm to mine*.

As shown in Figure 1, some miners in blockchain join a malicious mining pool for honest mining or FAW attacks on another mining pool to obtain mining revenue or FAW attack revenue and reduce the revenue of the target mining pool. During the mining process, the mining pool manager can receive PoW data reported by miners and know the number of network forks caused by the miners, the communication delay, and the current accumulated time of joining the mining pool.

Based on the behavior information of a large number of miners, the mining pool manager detects malicious miners, but the following four problems need to be solved: firstly, how to propose a miner data preprocessing method. The method needs to perform data cleaning and initial verification to quickly correct erroneous values in the data and initially screen out some malicious miners; secondly, how to propose a behavioral reward and punishment mechanism and a credit scoring model: they need to provide reward and punishment according to the PoW completed by miners, and comprehensively evaluate the block mining behavior of miners; thirdly, how to study the classification of credit levels, and divide miners into different credit levels by information such as mining behavior and credit value of miners, which can obtain the distribution of malicious miners in the mining pool, and reduces the revenue distribution of malicious miners; fourthly, how to quickly evaluate the effect of malicious miner protection by the target mining pool, which is convenient for the rapid simulation verification of the algorithm.

### 3.2. Basic Principles

Malicious miners are challenging to detect initially, but malicious miners carry out FAW attacks and show different behaviors as time goes on. Therefore, the mining pool manager can use the information to detect malicious miners after learning the behavior information of miners. As shown in Figure 2, when the network runs for a period of time, according to the miners’ behavior information, we propose a FAW attack protection algorithm (FAWPA) based on miners’ mining behavior in the blockchain. It mainly includes miner data preprocessing and malicious miner detection. The implementation scheme is as follows.

#### 3.2.1. Miner Data Preprocessing

After receiving the miners’ PoW, the number of network forks, communication delay, and offline times, the mining pool manager cleans and normalizes the information. Then it uses the boxplot analysis algorithm to detect the received data [19]. When the mining pool manager finds the wrong value, it replaces the incorrect value with the mean of the normal value. According to the processed information, the mining pool manager uses various verification methods such as data format, time validity, and data validity to complete the initial malicious detection of miners. Namely, it analyzes the data format, the chronological sequence of reported data and the number of error values, and counts the number of errors in the data. Then it carries out single-dimensional data preprocessing of miner data. If any of the detected miner’s error times and offline times is greater than threshold $th_{1}$, or the communication delay is higher than threshold $th_{2}$, then the mining pool manager gives the miner an *inferior* credit rating and the revenue weight $\mu_{2}$.

#### 3.2.2. Malicious Miner Detection

##### Behavioral Reward and Punishment Mechanism and Credit Value Scoring Model

Considering the specific behavior of miners, we propose a reward and punishment mechanism for miners’ behavior, and calculate the cumulative performance value of miners in block mining. Among them, the reward and punishment mechanism is mainly composed of two aspects, such as PoW reward and block mining failure punishment. PoW rewards are divided into two categories. In the first category, the miner successfully obtains the full PoW certificate, and its cumulative performance value increases by performance point α. In the second category, when the miner obtains a PpoW, its cumulative performance value increases by performance point β. The block mining failure punishment is divided into two categories. In the first category, when another mining pool successfully obtains the FPoW certificate, the cumulative performance value of all miners in this mining pool is deducted by performance point χ. In the second category, when the block submitted by the miner causes a network fork, the miner’s cumulative performance value is deducted by performance point ε. Then, FAWPA calculates the performance value $\delta_{i}$ of each miner *i* in the block mining.
(1)$$\delta_{i}=\phi_{i}\beta +\varphi_{i}\alpha -\gamma \chi -\eta \varepsilon$$
where $\phi_{i}$ represents the number of POW certificates of miner *i*, $\varphi_{i}$ represents the flag whether miner *i* has a FPOW certificate, γ represents the flag whether the mining pool with miner *i* has a FPOW certificate, η represents the flag whether the block submitted by miner *i* causes a network fork. Then, FAWPA calculates the cumulative performance value of miner *i*.
(2)$$\iota_{i}=\kappa {\iota^{\prime }}_{i}+\delta_{i}$$
where $\iota_{i}$ represents the updated cumulative performance value of miner *i* at the current moment, ${\iota^{\prime }}_{i}$ represents the cumulative performance value of miner *i* at the previous moment, and κ represents the update factor of the performance value.

After updating the cumulative performance value of miners, we propose the credit value scoring model of miners. The model mobilizes the enthusiasm of miners to participate in mining and serve as the basis for credit ratings.
(3)$${ο}_{i}=\frac{\lambda_{1}\times (\iota_{i}/\lambda_{2})}{1+\left|\iota_{i}/\lambda_{2}\right|}$$
where ${ο}_{i}$ represents the accumulated credit value of miner *I*, which is used as the evaluation factor for credit rating division, and $\lambda_{1}$ and $\lambda_{2}$ represent the model parameters.

##### Credit Rating Classification

According to the information collected in the data preprocessing stage, FAWPA obtains evaluation elements of the miner credit rating classification in Table 1, including current credit value, cumulative credit value, offline times, delay time, current network forks, and other information. Then we propose the miner credit rating classification according to the above evaluation elements. As shown in Formula (4), the miner credit rating value is defined as follows:(4)QR={superior,inferior}
where QR represents the credit rating set.

The traditional clustering algorithm has the following problems in the actual clustering process:

(1) The clustering algorithm based on sample membership represented by FCM (Fuzzy C-Means) focuses on optimizing the least square error. Namely, it optimizes the distance between the object in the cluster and the center of the cluster. However, it ignores the distance between different clusters, and partial clusters of discrete samples cannot be divided [20].

(2) The clustering algorithm based on sample distance represented by HC (Hierarchical Clustering) assigns the same weight to all features in the process of distance calculation. It does not consider that there are certain differences in the importance of different features. Therefore, the adaptability in different scene environments is poor [21].

(3) The clustering algorithm based on cluster center selection represented by K-means is easily affected by the initial cluster center selection. It results in unstable clustering results and makes it easy to fall into the optimal local solution [22].

Therefore, the clustering effect of existing algorithms is not ideal, and the running time of existing algorithms is too long. They cannot meet the precision rate and real-time requirements of credit rating classification [23]. The Aquila Optimizer (AO) algorithm [24] has a good search ability for global optimal solutions. The AO algorithm simulates Aquila’s high soar flight and contour flight to quickly determine the scope of the global optimal solution, and simulates the low flight and swoop by walk capture of Aquila to accurately search for the global optimal solution. However, in the optimization process, the AO algorithm mainly realizes different optimization links according to the value of random numbers, and cannot reach a convergence state in a short time. Therefore, we propose a miner credit rating classification mechanism (MCCM) based on AO in view of the above problems. MCCM considers the distance inside and outside the cluster, and improves the fitness calculation method. It also introduces multiple weight parameters to improve the distance calculation method. It simulates Aquila’s predation behavior to optimize the cluster center selection, and improves the solution update mechanism in different optimization stages. Then, it improves convergence speed and finally realizes the rapid classification of miners’ credit ratings. The details are as follows.

MCCM determines the maximum and minimum values of different dimensions based on the miners’ credit rating classification and evaluation elements, and constructs a search space. Then, it randomly generates *K* cluster center positions in the search space as a solution *x*, and groups the ζ solutions into a solution matrix X. Since the purpose of the clustering effect is to maximize the distance between different clusters while minimizing the distance within the cluster, we assume that all samples are clustered according to their distances from the centers of the cluster. When the position of the cluster center is reasonable, it can effectively reduce the distance within cluster and increase the distance between different clusters to a certain extent, and achieve a better clustering effect. Therefore, MCCM proposes the fitness of different solutions according to Formula (5), and selects the solution with the largest fitness as the best solution $x_{b}$.
(5)$$F_{z}=\frac{\sum_{l=1}^{K}\sum_{j=1}^{K}B(C_{l},C_{j})}{\sum_{l=1}^{K}\sum_{k=1}^{SN}D(x_{k,}{}_{l},C_{l})}$$
where $F_{z}$ represents the fitness value of the *z*th solution, *K* represents the number of clusters, SN represents the number of samples in the cluster, and $C_{l}$ and $C_{j}$ respectively represent the cluster center positions of the *i*th cluster and *j*th cluster obtained from the *z*th solution. $x_{k,l}$ represents the *k*th sample in the *l*th cluster. $D(x_{k,l},C_{l})$ represents the distance between the *k*th sample and the cluster center of the *l*th cluster. $B(C_{l},C_{j})$ represents the distance between the cluster center position of the *l*th cluster and the cluster center position of the *j*th cluster. In terms of calculating distance, MCCM considers the important degree of different evaluation elements and sets different weight parameters to clear differences between malicious miners and honest miners. The formula is as follows:(6)$$\begin{array}{cc} D(x_{k,}{}_{l},C_{l})= & \varpi_{1}\times \sqrt{{\left(\varepsilon_{k,l}^{1}-\varepsilon_{l}^{1}\right)}^{2}}+\varpi_{2}\times \sqrt{{\left(\varepsilon_{k,l}^{2}-\varepsilon_{l}^{2}\right)}^{2}}\\ & +\varpi_{3}\times \sqrt{{\left(\varepsilon_{k,l}^{3}-\varepsilon_{l}^{3}\right)}^{2}}+\varpi_{4}\times \sqrt{{\left(\varepsilon_{k,l}^{4}-\varepsilon_{l}^{4}\right)}^{2}}+\varpi_{5}\times \sqrt{{\left(\varepsilon_{k,l}^{5}-\varepsilon_{l}^{5}\right)}^{2}}\\ B(C_{l},C_{j})= & \varpi_{1}\times \sqrt{{\left(\varepsilon_{l}^{1}-\varepsilon_{j}^{1}\right)}^{2}}+\varpi_{2}\times \sqrt{{\left(\varepsilon_{l}^{2}-\varepsilon_{j}^{2}\right)}^{2}}\\ & +\varpi_{3}\times \sqrt{{\left(\varepsilon_{l}^{3}-\varepsilon_{j}^{3}\right)}^{2}}+\varpi_{4}\times \sqrt{{\left(\varepsilon_{l}^{4}-\varepsilon_{j}^{4}\right)}^{2}}+\varpi_{5}\times \sqrt{{\left(\varepsilon_{l}^{5}-\varepsilon_{j}^{5}\right)}^{2}} \end{array}$$
where $\varepsilon_{k,l}^{1}$ represents the current cumulative credit value of the *k*th sample in the *l*th cluster, $\varepsilon_{k,l}^{2}$ represents the historical rating value of the *k*th sample in the *l*th cluster, $\varepsilon_{k,l}^{3}$ represents the offline times of the *k*th sample in the *l*th cluster, $\varepsilon_{k,l}^{4}$ represents the delay time of the *k*th sample in the *l*th cluster, $\varepsilon_{k,l}^{5}$ represents the current number of network forks of the *k*th sample in the *l*th cluster, $\varepsilon_{l}^{1}$ and $\varepsilon_{j}^{1}$ respectively represent the current cumulative credit values of the cluster center positions in the *l*th cluster and the *j*th cluster, $\varepsilon_{l}^{2}$ and $\varepsilon_{j}^{2}$ respectively represent the historical level values of the cluster center positions in the *l*th cluster and the *j*th cluster, $\varepsilon_{l}^{3}$ and $\varepsilon_{j}^{3}$ respectively represent the offline times of the cluster center positions in the *l*th cluster and the *j*th cluster, $\varepsilon_{l}^{4}$ and $\varepsilon_{j}^{4}$ respectively represent the delay time of the cluster center position in the *l*th cluster and the *j*th cluster, $\varepsilon_{l}^{5}$ and $\varepsilon_{j}^{5}$ respectively represent the current number of network forks at the cluster center positions in the *l*th cluster and the *j*th cluster, $\varpi_{1}$ represents the weight value of the current accumulated credit value, $\varpi_{2}$ represents the weight value of the historical rating value, $\varpi_{3}$ represents the weight value of offline times, $\varpi_{4}$ represents the weight value of delay time, and $\varpi_{5}$ represents the weight value of the current number of network forks.

In different iterative processes, the solution needs to be adjusted appropriately, and the similarity ratio $\tau_{k}$ of the *k*th solution is calculated by the Formula (7).
(7)$$\tau_{k}=\sqrt{\sum_{j=1}^{\xi }{\left(x_{k,j}-{x^{\prime }}_{k,j}\right)}^{2}}$$
where $\tau_{k}$ represents the similarity ratio between the *k*th solution in the current iteration process and its previous iteration process, which is used to switch among different optimization stages, $x_{k,j}$ represents the *j*th miner credit rating evaluation element for the *k*th solution in the current iteration process, ${x^{\prime }}_{k,j}$ represents the *j*th miner credit rating evaluation element for the *k*th solution in the previous iteration process, ξ represents the dimension of a single solution. Maxit represents the maximum number of iterations, it represents the current iteration number, and $\upsilon_{1}$ and $\upsilon_{2}$ represent similarity thresholds for solution updates. The specific update of the solution is as follows:

(1) When it≤2×Maxit/3 and $\tau_{k}\geq \upsilon_{1}$, since the current solution is far away from the optimal solution, it is necessary to quickly determine the search range of the optimal solution by simulating the Aquila’s high soar flight by Formula (8). It can find the optimal cluster center position later.
(8)$$x^{\prime \prime }=x_{b}\times (1-it/Maxit)+(x_{c}-x_{b})\times r_{1}$$
where $x^{\prime \prime }$ represents the updated solution, $x_{c}$ represents a ξ×1 vector composed of the average values of solution x, $x_{b}$ represents the best solution in the current iteration process, and $r_{1}$ represents a random number in the range from zero to one.

(2) When it≤2×Maxit/3 and $\tau_{k}<\upsilon_{1}$, since the search range of the optimal solution is relatively wide, it is necessary to further narrow the current search range by simulating the Aquila’s outline flight by Formula (9). It can facilitate the rapid approach to the global optimal solution in the later stage.
(9)$$x^{\prime \prime }=x_{b}\times L+x_{d}+(e_{2}-e_{1})\times r_{1}$$
where L represents the random value of Lévy flight, $x_{d}$ represents a randomly selected solution in the solution matrix X, and $e_{1}$ and $e_{2}$ represent random value vectors that simulate Aquila spiral search.

(3) When it>2×Maxit/3 and $\tau_{k}\geq \upsilon_{2}$, the solution needs transit from large-scale search to small-scale optimization. Therefore, the solution uses Formula (10) to simulate the low flight of Aquila. Then it achieves a fast approach to the global optimal solution. It is convenient for the next step to accurately search for the optimal cluster center position.
(10)$$x^{\prime \prime }=(x_{b}-x_{e})\times \omega_{1}-r_{2}+((up-lp)\times r_{1}+lp)\times \omega_{2}$$
where $x_{e}$ represents the mean vector of solution matrix X in different dimensions, $\omega_{1}$ and $\omega_{2}$ represent solution search parameters in the range zero to one, $r_{2}$ represents a vector composed of random numbers in the range zero to one, up represents a ξ×1 vector composed of maximum values in different dimensions, and lp represents a ξ×1 vector composed of the minimum values in different dimensions.

(4) When it>2×Maxit/3 and $\tau_{k}<\upsilon_{2}$, the solution is close to the optimal solution. Therefore, the solution calculates the mass function to ensure an accurate search by Formula (11). Then, it combines Formula (12) to simulate the flight capture of Aquila. It can accurately search for the optimal cluster center position.
(11)$$f=it^{\frac{2\times r_{1}-1}{{\left(1-Maxit\right)}^{2}}}$$
(12)$$x^{\prime \prime }=f\times x_{b}-(g_{1}\times x\times r_{1})-g_{2}\times L+r_{2}\times g_{1}$$
where $g_{1}$ represents a random value in the process of simulating Aquila capturing prey and $g_{2}$ represents the flight slope of the simulated Aquila during prey capture. On the basis of obtaining the optimal solution of radius, MCCM calculates the evaluation value of each cluster by Formula (13), and divides the miners into *superior* and *inferior* levels. After rating classification is completed, the mining pool manager assigns revenue distribution weight $\mu_{1}$ to the miners with *superior* credit and assigns weight $\mu_{2}$ to the miners with *inferior* credit.
(13)$$c_{l}=\frac{\sum_{k=1}^{SN}\vartheta_{1}\times \varepsilon_{k,l}^{1}+\vartheta_{2}\times \varepsilon_{k,l}^{2}-\vartheta_{3}\times \varepsilon_{k,l}^{3}-\vartheta_{4}\times \varepsilon_{k,l}^{4}-\vartheta_{5}\times \varepsilon_{k,l}^{5}}{SN}$$
where $c_{l}$ represents the evaluation value of the *l*th cluster, $\vartheta_{1}$ represents the weight value of current accumulated credit value, $\vartheta_{2}$ represents the weight value of the historical *r* value, $\vartheta_{3}$ represents the weight value of offline times, $\vartheta_{4}$ represents the weight value of the delay time, and $\vartheta_{5}$ represents the weight value of the current number of network forks. After a certain period of time, the mining pool manager clears miners’ accumulated performance values and accumulated credit values, and starts to calculate the accumulated performance values again, so as to ensure the revenue of high-credit miners and reduce the revenue of malicious miners.

#### 3.2.3. Mining Pool Revenue Distribution and Evaluation

According to credit rating results, the mining pool manager defines different revenue weights for each different level, and redistributes the revenue of mining pool. $C_{T}$ represents the total computing power of target mining pool, $C_{O}$ represents the total computing power of other mining pools, $C_{E}$ represents the total computing power of malicious mining pools and $C_{A}$ represents the total computing power of the entire network, that is, $C_{E}+C_{O}+C_{T}=C_{A}$. The total computing power $C_{E}$ of the malicious mining pool consists of the computing power $C_{E}^{H}$ used for honest mining and the computing power $C_{E}^{B}$ used to carry out FAW attacks.
(14)$$C_{E}^{H}=\underset{i}{\sum }p_{i}h_{iE},C_{E}^{B}=\underset{i}{\sum }p_{i}d_{iE}$$
where $p_{i}$ represents the computing power of miner *i*, $h_{iE}$ represents the indicator that miner *i* carries out honest mining according to the assignment of the malicious mining pool, $d_{iE}$ represents the indicator whether miner *i* carries out a FAW attack on the target mining pool. The revenue of the malicious mining pool is mainly divided into the following three parts: First, the malicious mining pool’s own honest mining computing power $C_{E}^{H}$ carries out block mining to obtain honest mining revenue; second, the malicious mining pool assigns computing power $C_{E}^{B}$ to carry out block withholding attacks without providing full proof of work to the target mining pool. At that time, the target mining pool assigns its honest mining revenue to the attack computing power; third, the malicious mining pool assigns computing power $C_{E}^{B}$ to carry out selfish mining attacks immediately when other mining pools find valid blocks. Namely, it announces the blocks that were discovered before and achieves a network fork. Then, the probability that the attack computing power allows the target mining pool to obtain block revenue is $v\times \sum_{i}p_{i}d_{iE}\times C_{O}/(C_{A}-\sum_{i}p_{i}d_{iE})$, where v represents the probability that the block submitted by the malicious miner is successfully uploaded to the chain. The target mining pool assigns its revenue to the attack computing power. Therefore, the total revenue of the malicious mining pool is
(15)$$R_{E}=\frac{\sum_{i}p_{i}h_{iE}}{C_{A}-\sum_{i}p_{i}d_{iE}}+\frac{C_{T}}{C_{A}-\sum_{i}p_{i}d_{iE}}\times \frac{\sum_{i}p_{i}d_{iE}}{C_{T}+\sum_{i}p_{i}d_{iE}}+v\times \frac{\sum_{i}p_{i}d_{iE}}{C_{A}}\times \frac{C_{O}}{C_{A}-\sum_{i}p_{i}d_{iE}}\times \frac{\sum_{i}p_{i}d_{iE}u_{i}}{C_{T}+\sum_{i}p_{i}d_{iE}}$$
where $u_{i}$ represents the weight factor of miner *i* in target mining pool. The weight factor $u_{i}$ of miner *i* in target mining pool that can detect malicious miners is
(16)$$u_{i}=\left\{ \begin{array}{l} \mu_{1},\mathrm{malicious}\;\mathrm{miner}\;i\;\mathrm{in}\;\mathrm{target}\;\mathrm{mining}\;\mathrm{pool},\;\mathrm{and}\;\mathrm{the}\;\mathrm{grade}\;\mathrm{is}\;\mathrm{superior}\\ \mu_{2},\mathrm{malicious}\;\mathrm{miner}\;i\;\mathrm{in}\;\mathrm{target}\;\mathrm{mining}\;\mathrm{pool},\;\mathrm{and}\;\mathrm{the}\;\mathrm{grade}\;\mathrm{is}\;\mathrm{inferior} \end{array}\right.$$
where $\mu_{1}$ and $\mu_{2}$ represent the revenue distribution weight. The revenue distribution weights $\mu_{1}$ and $\mu_{2}$ of miner *i* in the target mining pool *T*, which does not detect the malicious miners, are both one. The target mining pool can still obtain honest mining revenue and network fork revenue under the FAW attack of the malicious mining pool, so the total revenue of the target mining pool is
(17)$$R_{T}=\frac{C_{T}}{C_{A}-\sum_{i}p_{i}d_{iE}}\times \frac{\sum_{i}p_{i}T_{ih}u_{i}}{C_{T}+\sum_{i}p_{i}d_{iE}}+v\times \frac{\sum_{i}\left(p_{i}d_{iE}\right)}{C_{A}}\times \frac{C_{O}}{C_{A}-\sum_{i}p_{i}d_{iE}}\times \frac{\sum_{i}p_{i}T_{ih}u_{i}}{C_{T}+\sum_{i}p_{i}d_{iE}}$$
where, $T_{ih}$ represents the flag that miner *i* carries out honest mining according to the assignment of the honest mining pool.

Other mining pools always obtain revenue through honest mining, that is,
(18)$$R_{O}=\frac{C_{O}}{C_{A}-\sum_{i}p_{i}d_{iE}}$$

Since miners in the mining pools need to consume resources such as electricity, water, and equipment damage when they carry out honest mining and FAW attacks [25], therefore, $t_{H}$ is the cost of honest mining consumption per unit of computing power, $t_{F}$ is the cost of FAW attack per unit of computing power. The net revenue in three types of mining pools is the total revenue minus the cost of computing power. The net revenue models are
(19)$${R^{\prime }}_{E}=\frac{R_{E}-C_{E}^{H}\times t_{H}-C_{E}^{B}\times t_{F}}{N_{E}},\;{R^{\prime }}_{T}=\frac{R_{T}-C_{T}\times t_{H}}{N_{T}},\;{R^{\prime }}_{O}=\frac{R_{O}-C_{O}\times t_{H}}{N_{O}}$$
where ${R^{\prime }}_{E}$ represents the net revenue of a miner in the malicious mining pool, ${R^{\prime }}_{T}$ represents the net revenue of a miner in the target mining pool, ${R^{\prime }}_{O}$ represents the net revenue of a miner in other mining pools, $N_{E}$ represents the number of miners that malicious mining pool can assign, $N_{T}$ represents the number of miners that target mining pool can assign, and $N_{O}$ represents the number of miners that other mining pools can assign.

## 4. Algorithm Implementation

The mining pool manager executes FAWPA to detect malicious miners, thereby increasing its revenue. The specific pseudo-code of FAWPA is shown in Algorithm 1: In line 1, the mining pool manager initializes parameters, such as the number of miners in the entire network n, the type of mining pools in the entire network pool, the computing power ratio of malicious mining pool attack κ, the step size control parameter lstep, the time threshold T, the credit value of the whole network miners οi, and starts timing. In lines 3–8, the mining pool manager collects the proof of work, offline times, delay time, current network forks, and other behavioral information of all miners in the mining pool; then, it uses the box plot detection algorithm to clean the data reported by miners and finds the wrong value. Then, it replaces it with the mean value of the normal value. According to the preprocessed data, the mining pool manager carries out preliminary verification. Namely, if any of the information, such as the number of errors and the number of offline times reported by miners, is greater than threshold th1, or the communication delay time is higher than threshold th2, it determines the credit rating as *inferior* directly. In lines 9–10, according to the previous work of miners, the mining pool manager implements a reward and punishment mechanism, and updates the cumulative performance value of each miner. At the same time, the mining pool manager converts the cumulative performance value into cumulative credit value with the credit value scoring model by Formula (3). In line 11, according to the data reported by miners, the mining pool manager obtains the evaluation elements shown in Table 1. It uses MCCM to achieve credit rating classification [24], and calculates the evaluation values of different clusters by Formula (13). Then it divides the miners into different credit ratings. In line 12, according to miners’ credit rating results, the mining pool manager sets miners’ revenue distribution weights. In lines 13–15, after a period of time, the mining pool manager clears the cumulative performance value and cumulative credit value of all miners, and starts a new round of cumulative performance value acquisition. In lines 17–18, the mining pool manager judges the status of the mining pool. If the mining pool successfully mines the block, the mining pool manager will assign the current round of mining pool revenue with the revenue distribution weight of each miner. In line 20, if other mining pool blocks are not successfully mined, the algorithm returns to data reception and preprocessing steps. Otherwise, the algorithm performs mining pool revenue distribution. In line 22, the network synchronizes the block information and mines the next block. The mining pool manager repeats the above steps (lines 3–23) to detect malicious miners who carry out FAW attacks. Then it reduces the weights of their revenue distribution and realizes the security protection of FAW attacks.
**Algorithm 1:** FAW Attack Protection Algorithm Based on the Behavior of Blockchain Miners (FAWPA)Input: behavior information of miners in target mining pool1: pool = 3; n = 1000; *lstep* = 1.5; T = 30; οi = 60; The mining pool manager starts timing;2: **while**(1)3:    The mining pool manager of the target mining pool receives the behavior information of each miner;4:    **if** the target mining pool knows the number of mines and blocks mined by any mining pool in the network, **then**5:         The target mining pool selects the behavior data of miners when the block is mined;6:         **if** miner behavior data exceeds the threshold of the boxplot analysis algorithm, **then**7:            The target pool managers perform data preprocessing and flag malicious miners;8:         **end**9:    The target pool manager implements a reward and punishment mechanism;10:   The target pool manager implements the credit value scoring model and updates the cumulative credit value οi of each miner;11:   The target pool manager implements the credit rating mechanism MCCM to classify miners’ credit ratings;12:   The target pool manager sets revenue distribution weight according to credit rating result;13:      **if**
*time* > *T*14:        The target pool manager clears cumulative performance value and cumulative credit value for all miners, time=0;15:      **end**16:   **end**17:   **if** the target pool manager successfully mines blocks, **then**18:      The target pool manager distributes mining pool revenue according to the revenue distribution weight of each miner;19:   **else**
20:       Return to line 3;21:   **end**22:   The network synchronizes the block information and starts to mine the next block;23: **end**

## 5. Experimental Simulation

### 5.1. Simulation Parameters and Performance Parameters

According to the statistics from the btc.com website on 19 November 2021, the computing power distribution of the mining pool in the Bitcoin network is shown in Table 2. Referring to the computing power distribution of the mining pool in the Bitcoin network, we set up the following simulation experiment scenario. We simulate the F2Pool and SlushPool to carry out a FAW attack on the Foundry USA mining pool, and the remaining mining pools as other mining pools. The attack mining pool selects a number κ of its own miners to carry out FAW attacks on all target mining pools, and the remaining miners carry out honest mining. Target mining pools can implement a protection algorithm. It can detect malicious miners and distribute the corresponding revenue. The target mining pool and other mining pools perform honest mining.

FAWPA detects malicious miners based on the behavior data of miners. Therefore, FAWPA can work with private chains and public chains based on the POW consensus mechanism, and can also work with a consortium blockchain. To verify the performance of FAWPA, according to the above simulation experiment scenario, we selected Ethereum and built a blockchain prototype system [26,27] to obtain the behavior data set of each miner *i* under FAW attack. The system uses Golang language on a server with Intel i7-10700 CPU 2.90GHz and 16G memory. Its operating system is 64-bit Win10. We also used the simulation parameters in Table 3 and used Matlab R2018b software to analyze the influence of credit value model parameters, malicious miner detection parameters, and block mining number on the precision rate of the algorithm. Then, considering the ratio of the number of mined blocks to malicious computing power, we calculated the following parameters of the target mining pool, such as cumulative revenue of computing power, and the recall rate and precision rate of malicious miners. 

Finally, when the malicious computing power changes, we compared the revenue of the target mining pools which use FAWPA, ND (No Detection), RSCM [13], AWRS [14], and ICRDS [16], where ND represents a target mining pool that does not detect the malicious miners. The recall rate is defined as the percentage of the number of real detected malicious miners to the total number of real malicious miners in the network. The precision rate is defined as the percentage of the number of real detected malicious miners to the total number of malicious miners deemed by the algorithm.5.2. Simulation Parameters and Performance Parameters

### 5.2. Simulation Analysis

#### 5.2.1. Miner Data Preprocessing

First, we selected the credit value model parameter $\lambda_{1}$ as 10, 20, 30, 40, 50, and 60, the model parameter $\lambda_{2}$ as 100, 200, 300, 400, 500, 600, and 700, the number of malicious miners κ as 10, 40, 70, 100, and 130, and other parameters in Table 3. Then, we analyzed the influence of credit value model parameters on the precision rate of malicious miners and the average non-malicious computing power revenue of the target mining pool. Then, we took κ=100 as an example to illustrate the influence of credit value model parameters on FAWPA.

As shown in Figure 3, when the model parameter $\lambda_{1}$ is 50 and the model parameter $\lambda_{2}$ is 500, the precision rate of malicious miners in the target mining pool reaches the maximum value. The reason is that when parameter $\lambda_{1}$ changes from 10 to 50, the range of the cumulative credit value of all miners in the model increases. The scattered credit values of malicious miners and honest miners are more, and the clustering and credit rating evaluation effect of FAWPA becomes better. Therefore, its precision rate increases accordingly. When parameter $\lambda_{2}$ changes from 100 to 500, the increase rate of miners in cumulative credit value slows down, and the cumulative credit value can reflect the situation of participating in block mining. FAWPA distinguishes malicious miners more accurately, so its precision rate increases accordingly. However, when parameter $\lambda_{1}$ reaches 60 or parameter $\lambda_{2}$ reaches 600, the cumulative credit value interval in the model is too large, or the growth rate of miners in the cumulative credit value is too slow. Therefore, the detection effect of some malicious miners deteriorates, and the precision rate of the target mining pool does not improve or even decreases slightly.

As shown in Figure 4, when the model parameter $\lambda_{1}$ is 50 and the model parameter $\lambda_{2}$ is 500, the average revenue of the target mining pool reaches the maximum value. The reason is that when parameter $\lambda_{1}$ changes from 10 to 50 or parameter $\lambda_{2}$ changes from 100 to 500, the effect of FAWPA in detecting malicious miners becomes better, and its precision rate increases. As a result, FAWPA gives more malicious miners in the target mining pools lower revenue distribution weights and distributes more revenue to honest miners. Their average revenue increases accordingly. However, when parameter $\lambda_{1}$ reaches 60 or parameter $\lambda_{2}$ reaches 600, the precision rate of malicious miners in the target mining pool changes little, but the detection effect of malicious miners worsens. FAWPA mistakes more honest mining pools for malicious miners, and its recall rate becomes worse. Therefore, the average revenue of the target mining pool decreases. In summary, FAWPA chooses the credit model parameters $\lambda_{1}$ as 50 and $\lambda_{2}$ as 500. It can better distinguish malicious miners from honest miners and improve the average revenue of the target mining pool.

We selected the similarity threshold parameter $\upsilon_{1}$ of the selected solution as 0.3, 0.4, 0.5, 0.6, and 0.7, and the similarity threshold parameter $\upsilon_{2}$ as 0.3, 0.4, 0.5, 0.6, and 0.7, the number of malicious miners κ as 10, 40, 70, 100, and 130 and other parameters in Table 2. Then, we analyzed the influence of similarity threshold υ on fitness value, and took κ=100 as an example to illustrate the influence. As shown in Figure 5, when the malicious detection parameter $\upsilon_{1}$ is 0.5 and the malicious detection parameter $\upsilon_{2}$ is 0.5, the fitness value reaches the maximum value. The reason is that when parameters $\upsilon_{1}$ and $\upsilon_{2}$ change from 0.5 to 0.3, the similarity threshold range gradually decreases. It leads to reducing the Aquila search range, and the found solution falls into a local optimal solution. Therefore, the fitness value decreases accordingly. When the parameters $\upsilon_{1}$ and $\upsilon_{2}$ change from 0.5 to 0.7, the range of the similarity threshold gradually increases. It leads to the extension of the Aquila search range, and it is difficult to find the global optimal solution quickly and accurately. Therefore, the fitness value also decreases. When both parameters $\upsilon_{1}$ and $\upsilon_{2}$ are set to 0.5, the fitness value reaches the maximum value. Namely, FAWPA minimizes the distance between clusters and maximizes the distance between different clusters. Then, FAWPA achieves the optimal clustering effect of miners. Therefore, FAWPA chooses the parameters $\upsilon_{1}$ and $\upsilon_{2}$ as 0.5.

We selected the step size control parameters lstep as 0.5, 1.0, 1.5, and 2.0, the maximum number of iterations as 1000, the number of malicious miners κ as 10, 40, 70, 100, and 130, and other parameters in Table 2. Then, we analyzed the influence of the step size control parameters on the fitness value, and took κ=100 as an example to illustrate the influence of parameter lstep on the convergence rate of the global optimal solution. As shown in Figure 6, when the parameter lstep is 1.5, the global optimal solution has the fastest convergence rate. The reason is that when the step size control parameter is less than 1.5, the unit distance of the Aquila search range is too small, and the search speed for the optimal solution is slow. Therefore, the convergence rate is slow. When the parameter lstep is greater than 1.5, the unit distance of the Aquila search range is too large, and the number of searches in the same range is small. The local search ability is not enough, and the optimal solution search is relatively weak. Therefore, the number of iterations required to find the global optimal solution is the largest. When the lstep is 1.5, FAWPA quickly locks the Aquila search range in the early flight, and quickly finds the optimal solution in the later search process. The number of iterations required to find global optimal solution is minimum. Therefore, FAWPA chooses the parameter lstep as 1.5, so it can realize the fast search for the global optimal solution and has a high convergence rate.

#### 5.2.2. Performance Analysis

We selected the number of malicious miners κ as 10, 40, 70, 100, and 130, the number of blocks as 30, 40, 50, 60, 70, 80, 90, and 100, and other parameters in Table 2. Then, we analyzed the influence of the number of malicious miners κ and the number of mined blocks on the precision rate. As shown in Figure 7, no matter how the number of malicious miners κ changes, with the increase of the number of mined blocks, the precision rate of FAWPA in detecting malicious miners gradually increases. The precision rate reaches more than 97.5%. The reason is that when the number of blocks is small, the mining pool manager collects less information, such as the current credit value, cumulative credit value, offline times, delay time, and current network forks. Some malicious miners do not show malicious behavior, resulting in a low precision rate of malicious miners in FAWPA. As the number of blocks increases, malicious miners show more malicious behaviors than normal miners. FAWPA increases the data difference between malicious miners and normal miners through the reward and punishment mechanism and credit scoring model, and uses multiple attack characteristics as clustering dimensions. It makes miner credit classification based on eagle predation behavior more accurate. In the end, no matter how the number of malicious miners changes, FAWPA detects malicious miners accurately, so the precision rate of malicious miners gradually increases. When the number of blocks is greater than 90, the mining pool manager has collected enough information on the behavior of miners, and the malicious miners have shown obvious differences. Therefore, no matter how the number of malicious miners changes, the precision rate of malicious miners reaches more than 97.5%.

We selected the number of malicious miners κ as 10, 40, 70, 100, and 130, the number of mined blocks as 30, 40, 50, 60, 70, 80, 90, and 100, and other parameters in Table 3. Then, we analyzed the influence of the number of malicious miners and the number of mined blocks on the recall rate. As shown in Figure 8, no matter how the number of malicious miners changes, as the number of mined blocks increases, the recall rate of FAWPA for malicious miner detection gradually increases, and its recall rate reaches more than 95%. The reason is that when the number of blocks is small, the mining pool manager collects less miner behavior data, and some malicious miners do not show malicious behavior, so the recall rate is low. As the number of blocks increases, FAWPA first performs box plot detection, excludes malicious miners with obvious differences, and realizes preliminary data cleaning and verification. Then, FAWPA detects the remaining malicious miners through MCCM clustering. In the end, no matter how the number of malicious miners changes, FAWPA detects malicious miners comprehensively, so the recall rate gradually increases. When the number of blocks is greater than 90, the mining pool manager has collected enough information on the differential behavior of malicious miners and honest miners, so the recall rate of malicious miners is more than 95%. Among them, some malicious miners have never mined blocks and have not carried out FAW attacks. Their behaviors do not show obvious differences from honest mining behaviors. Therefore, FAWPA cannot detect these miners and the recall rate is slightly lower than the precision rate.

#### 5.2.3. Performance Comparison

We selected the number of malicious miners as 10, 40, 70, 100, and 130 and other parameters in Table 3. Then, we calculated the cumulative revenues of the target mining pool in ND, RSCM, AWRS, ICRDS, and FAWPA. As shown in Figure 9, the cumulative revenue of the target mining pool with FAWPA is greater than that with ND, RSCM, AWRS, and ICRDS. As the number of malicious miners increases, the cumulative revenue of the target mining pool with FAWPA increases, while the cumulative revenues of the target mining pool with ND, RSCM, AWRS, and ICRDS decrease accordingly. The reason is that FAWPA combines the multi-dimensional behavior information of miners, and evaluates the behavior of miners through reward and punishment mechanisms. Then, FAWPA uses a miner credit classification mechanism based on eagle predation behavior for miners. FAWPA can improve the precision rate and recall rate of malicious miner detection and assign different revenue weights to different types of miners. Therefore, it can effectively prevent FAW attacks from malicious miners. It also protects and improves the accumulated revenue of the target mining pool. AWRS eliminates FAW attack motivation of malicious miners by rewarding miners for workload. RSCM only evaluates the credit value of miners through the number of network forks caused by miners, and assists the mining pool manager in a time to eliminate malicious miners who carry out FAW attacks. ICRDS implements a reward and punishment mechanism based on the number of block submissions to limit malicious miners. AWRS, RSCM, and ICRDS do not fully consider the multi-dimensional behavior of malicious miners, while ND does not consider the detection of malicious miners, and its target mining pool has the lowest cumulative revenue. Therefore, the target mining pool with FAWPA has the highest cumulative revenue, greater than the cumulative revenues of the target mining pool with ND, RSCM, AWRS, and ICRDS. At the same time, with the increasing number of malicious miners, the behavior data of malicious miners and honest miners in the network are obviously different. The precision rate and recall rate of FAWPA in detecting malicious miners increases, resulting in a revenue decrease for malicious miners. 

We assumed that the total computing power of the network is constant in the experimental simulation. Therefore, the mining revenue of the honest mining computing power in the target mining pool increased accordingly. FAWPA increased the cumulative revenue of the target mining pool. However, ND, RSCM, AWRS, and ICRDS were less effective in defending against FAW attacks. They difficultly coped with the increasing number of malicious miners, and the cumulative revenues of target mining pools decreased accordingly.

## 6. Conclusions

This paper proposes a FAW attack protection algorithm (FAWPA) based on the behavior of blockchain miners. Firstly, FAWPA performs data cleaning and preliminary verification on the behavior of malicious mining pools that carry out FAW attacks on target mining pools. We proposed a behavior reward and punishment mechanism and a credit rating model to evaluate the block mining behavior of miners comprehensively. According to the multiple attack characteristics of miner behavior data, we proposed a miner’s credit classification mechanism based on fuzzy C-means (FCM), which combines the improved Aquila optimizer (AO). The mechanism can accurately and efficiently detect miners with malicious behavior and classify malicious miners in the mining pool with low credit levels. Then, it gives lower revenue distribution weights to protect and improve the revenue of the target mining pool. Secondly, we proposed a revenue model for the mining pool and a revenue model for each miner under FAW attack. Finally, we analyzed the influence of parameter selection on FAWPA and analyze the precision rate and recall rate of FAWPA. Then, we compared the algorithm performance of ND, RSCM, AWRS, and ICRDS.

The simulation results show that no matter how the number of malicious miners changes, FAWPA can comprehensively and efficiently detect malicious miners in the target mining pool. FAWPA also improves the recall rate and precision rate of malicious miner detection and improves the cumulative revenue of the target mining pool. However, FAWPA mainly considers the attack and protection between a malicious attack mining pool and a target mining pool, and does not consider the revenue problem among multiple malicious attacks and multiple target mining pools. Therefore, the next research goal is to study the protection algorithm among multiple mining pools under FAW attacks.

## Figures and Tables

**Figure 1 sensors-22-05032-f001:**
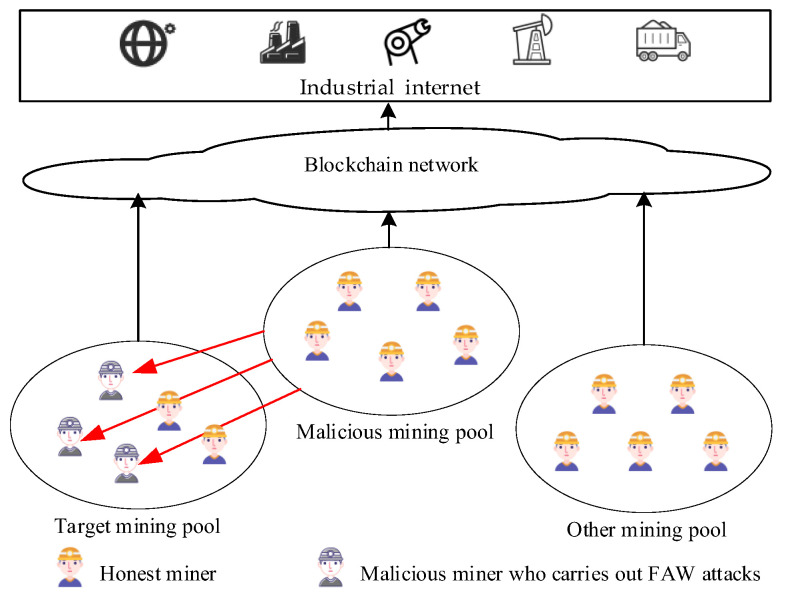
FAW attack diagram.

**Figure 2 sensors-22-05032-f002:**
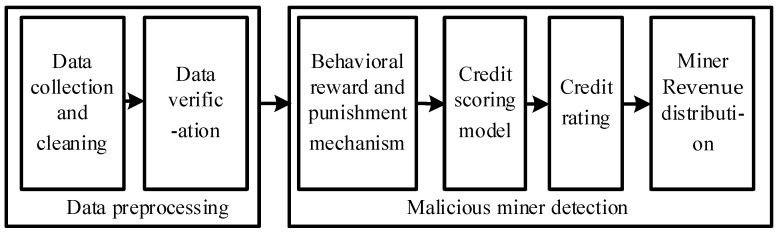
FAWPA schematic Diagram.

**Figure 3 sensors-22-05032-f003:**
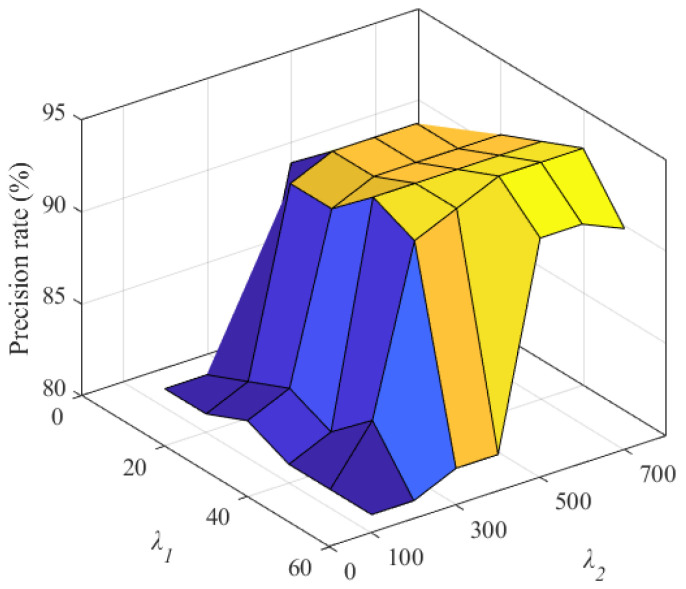
The influence of credit model parameters on precision rate.

**Figure 4 sensors-22-05032-f004:**
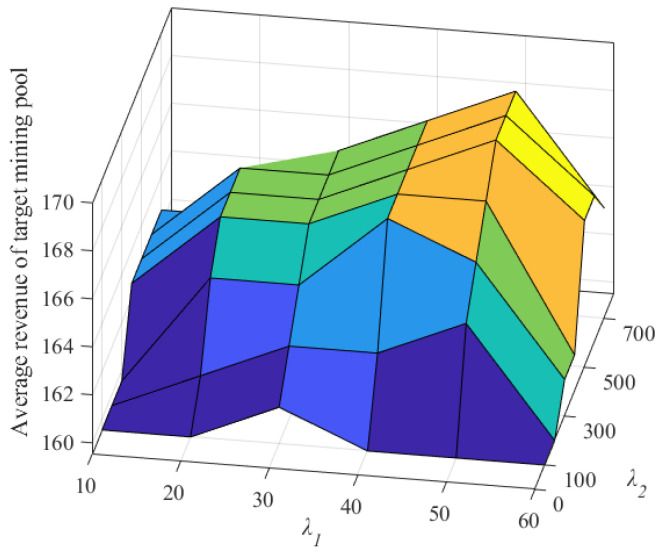
The influence of credit value model parameters on average revenue.

**Figure 5 sensors-22-05032-f005:**
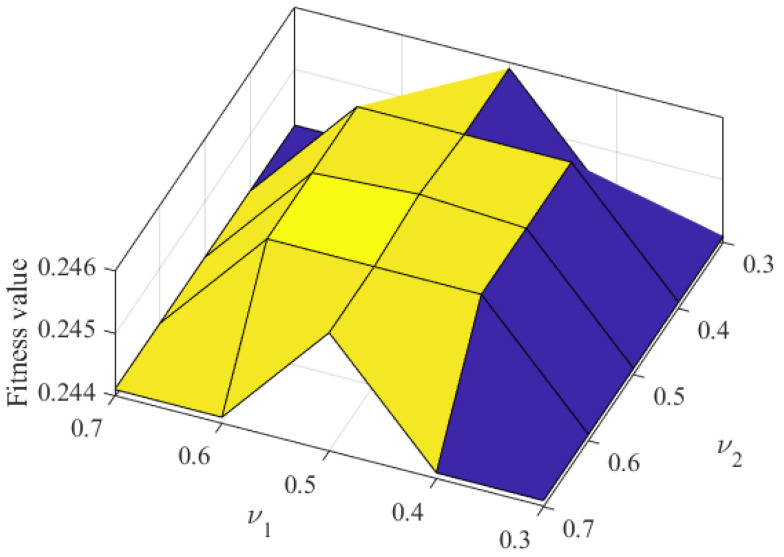
The impact of similarity threshold υ on fitness value.

**Figure 6 sensors-22-05032-f006:**
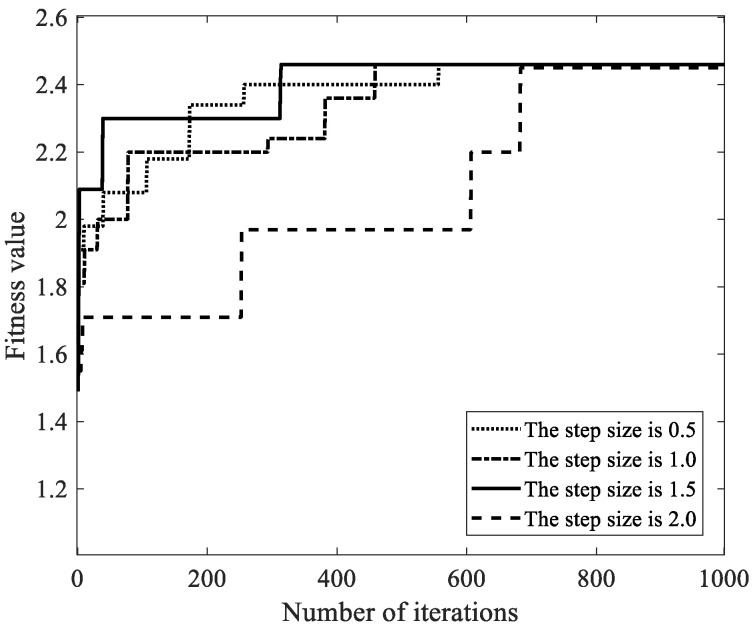
The influence of step length control parameter lstep on fitness value.

**Figure 7 sensors-22-05032-f007:**
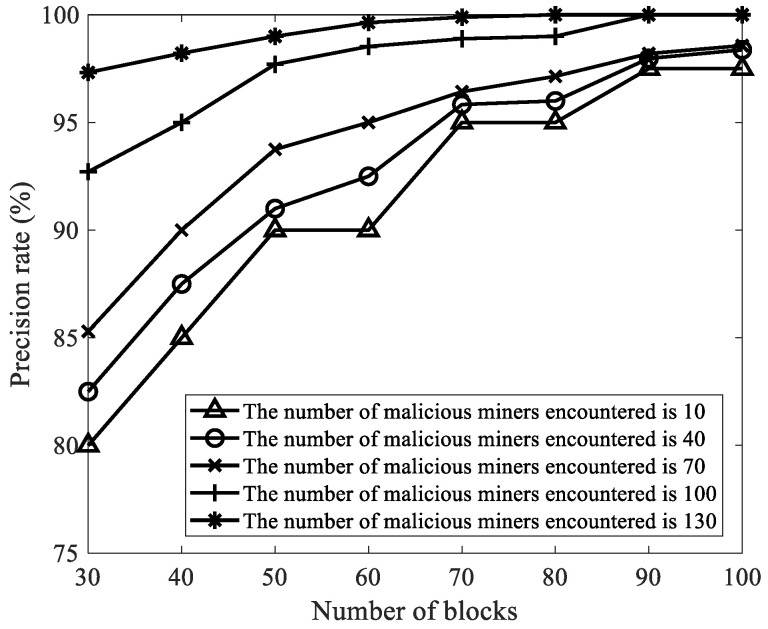
The influence of the number of malicious miners in the target mining pool on the precision rate.

**Figure 8 sensors-22-05032-f008:**
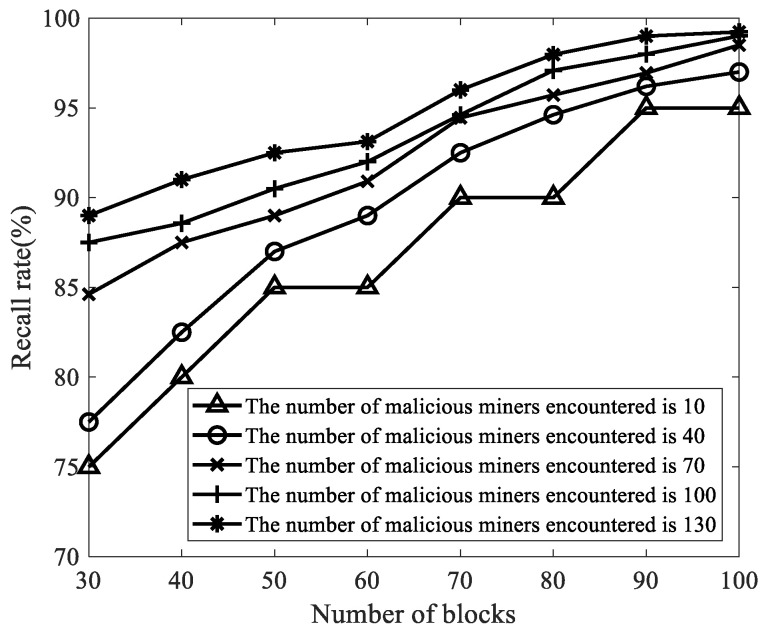
The influence of the number of malicious miners in the target mining pool on the recall rate.

**Figure 9 sensors-22-05032-f009:**
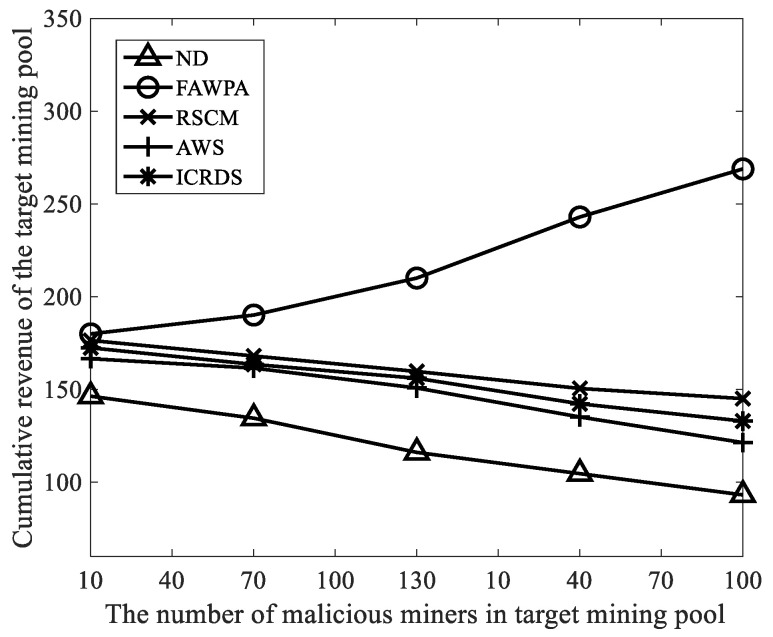
Cumulative revenue comparison of target mining pools.

**Table 1 sensors-22-05032-t001:** Evaluation element table of miner credit rating classification.

Name	Explanation
Current credit value	Miner’s current credit value for working in the block
Cumulative credit value	Miner’s cumulative credit value before credit rating classification
Offline times	Number of miner’s offline in the current mining pool
Delay time	Delay time of miner in communication process
Current number of network forks	Number of network forks caused by miner before credit rating classification

**Table 2 sensors-22-05032-t002:** The computing power of mining pool in the real Bitcoin network.

Rankings	Mining Pool	Computing Power	Proportion
1	Foundry USA	48,209.44 PH/s	21.51%
2	F2Pool	32,305.30 PH/s	14.41%
3	Binance Pool	30,814.28 PH/s	13.75%
4	Poolin	30,317.28 PH/s	13.53%
5	AntPool	21,371.20 PH/s	9.53%
6	ViaBTC	20,377.19 PH/s	9.09%
7	SlushPool	10,934.10 PH/s	4.88%
8	btc.com	10,437.10 PH/s	4.66%
9	SBI Crypto	6461.06 PH/s	2.88%
10	Luxor	5467.05 PH/s	2.44%
11	unknown	2982.03 PH/s	1.33%
12	MARA Pool	2982.03 PH/s	1.33%
13	Others	1490.99 PH/s	0.66%

**Table 3 sensors-22-05032-t003:** Simulation parameter table.

Parameter	Number	Parameter	Number
Number of miners	1000	Similarity threshold $\upsilon_{1}$	0.5
Number of blocks	100	Similarity threshold $\upsilon_{2}$	0.5
Number of cluster centers K	2	Credit model parameter $\lambda_{1}$	50
Initial credit value ${ο}_{i}$	60	Credit model parameter $\lambda_{2}$	500
Weight value of current credit Value $\varpi_{1}$	5	Weight value of offline times $\varpi_{3}$	10
Cumulative credit value $\varpi_{2}$	5	Weight value of delay times $\varpi_{4}$	10
Network fork punishment ε	50	Weight value of fork times $\varpi_{5}$	10
Block mining failure punishment χ	30	Proportion of computing power per unit of miners to the computing power of the entire network	0.1%
Performance value of FPoW reward α	200	Performance value of PPoW reward β	30
Honest mining cost per unit of computing $t_{H}$	10^−3^	FAW attack cost per unit of computing power $t_{F}$	10^−4^

## Data Availability

Not applicable.

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
