# Peer review of "FAWPA: A FAW Attack Protection Algorithm Based on the Behavior of Blockchain Miners"

_sensors, 2022, doi:10.3390/s22135032_

Round 1
Reviewer 1 Report
1. Use of undisclosed abbreviation FAW in the abstract is not advised.
2. The abstract is too long.
3. What is “combing”?
4. “such as block with-holding attack(BWH), selfish mining attack, and FAW (Fork after Withholding) attack [5].” [5] has nothing to do with FAW. Wrong reference.
5. “FAW attack is a new attack method combining block withholding attack and selfish mining attack [8].” We can find several definitions of FAW attack in [8]. However, the definition we have met firstly in the paper clearer reveals the essence of the FAW attack.
6. “FAW attack is a more threatening mining pool attack method.” in comparison with?
7. “The current FAW attack protection algorithms mainly have the following three problems.”. How can you enumerate problems of the current FAW attack protection algorithms, if they were not presented yet?
8. “related references mainly focus on improving the revenue of FAW attack [9-10],”. FAW attack is not considered in [9]. Wrong reference.
9. “The specific pseudo-code is as follows.”. Why no reference to the Algorithm?
10. “Figure8” – space is missing. The same is true for the other figures following Figure 8.
11. “proposes a FAW attack protection algorithm”. So, do the authors propose the FAW attack protection algorithm or do the authors propose the defense algorithm against FAW attack?
Author Response
Dear Reviewer,
We greatly appreciate the reviewers’ efforts to carefully review the paper and the valuable suggestions. Enclosed, please find the first round revised manuscript “FAWPA: A FAW Attack Protection Algorithm Based on the Behavior of Blockchain Miners”, which was submitted to the Editorial Office. As suggested by the reviewers, we have revised the paper in accordance with all the concrete comments and believe that the first revised version could meet the journal publication requirements.
We are uploading (a) our point-by-point response to the comments (below) (response to reviewers), (b) an updated manuscript with red words indicating changes, and (c) a clean updated manuscript without highlights.
For the convenience of checking our revised manuscript, our replies to each comment are as follows.
Best regards,
Yourong Chen et al.

Reviewer 2 Report
Dear Authors, This is an interesting topic. I have some comments which needs justification. As you have claimed that you have proposed a FAW attack protection algorithm to prevent the FAW attack in blockchain under the industrial internet based on the behavior of blockchain miners. Could you please give any example where you have tested this? how efficient your algorithm is and what type of blockchains it can work with. I can see lots of equation, it would be great to include explanation for those equations.
Author Response

(The authors gave the same response as above.)

Round 2
Reviewer 1 Report
Thanks to Authors. They have reacted to all my concerns.
Reviewer 2 Report
Dear Authors,
Thank you for providing the revised version of the manuscript. I am happy with the revisions.